# Essential Oils as a Source of Ecofriendly Insecticides for *Drosophila suzukii* (Diptera: Drosophilidae) and Their Potential Non-Target Effects

**DOI:** 10.3390/molecules27196215

**Published:** 2022-09-21

**Authors:** Michele Trombin de Souza, Mireli Trombin de Souza, Maíra Chagas Morais, Daiana da Costa Oliveira, Douglas José de Melo, Leonardo Figueiredo, Paulo Henrique Gorgatti Zarbin, Maria Aparecida Cassilha Zawadneak, Daniel Bernardi

**Affiliations:** 1Department of Plant Health, Federal University of Pelotas, Mailbox 354, Capão-do-Leão 96010-900, Rio Grande do Sul, Brazil; 2Department of Basic Pathology, Federal University of Parana, Mailbox 19031, Curitiba 81531-980, Parana, Brazil; 3Department of Chemistry, Federal University of Parana, Mailbox 19020, Curitiba 81531-990, Parana, Brazil

**Keywords:** spotted wing drosophila, *Trichopria anastrephae*, biopesticides, sustainable pest management

## Abstract

The spotted wing drosophila (*Drosophila suzukii*) is one of the main invasive pests of small fruits in the world. Thus, 19 essential oils (EOs) were selected to analyze the effects through toxicity and repellency on oviposition and *D. suzukii* adults. In addition, their lethal and sublethal effects on the pupal endoparasitoid *Trichopria anastrephae* were evaluated. The EOs of *C. flexuosus* and *Mentha* spp. had the highest toxicity observed in the topical application bioassay for *D. suzukii*. In contrast, the EOs of *C. verum*, *C. citratus* QT *citratus*, and *C. winterianus* showed the highest toxicity in the ingestion bioassay for *D. suzukii*. The dry residues of *C. verum* and *C. citratus* QT *citratus* reduced the oviposition of *D. suzukii*. In the repellency bioassays, the 19 EOs analyzed repelled ≅ 90% of the *D. suzukii* females. All EOs evaluated using the LC_90_ values of the products provided mortality of less than 20% of *T. anastrephae* adults and did not cause a reduction in the parasitism of surviving *T. anastrephae* females. We conclude that the EOs evaluated have the potential to be used in the management of *D. suzukii*. They can also serve as selective active ingredients for the formulation and synthesis of new biopesticides.

## 1. Introduction

The spotted wing drosophila *Drosophila suzukii* (Matsumura) (Diptera: Drosophilidae) is one of the main pests in the production of fruits with thin skins worldwide [1,2,3]. The *D. suzukii* female has a serrated ovipositor that is rarely found in other drosophilids [4]. This fact allows the species to lay eggs on ripening whole fruits. Subsequently, the larvae open galleries and promote accelerated decomposition of these fruits. The presence of *D. suzukii* represents a phytosanitary challenge due to its wide range of hosts, fostered by the absence of an economic threshold for this pest in fruit damage [5,6], and its occurrence in forests and marginal areas [7]. There are ≅ 80 host plants of *D. suzukii* [8], and their economic impacts have been documented in the northern [9,10] and southern hemisphere [11].

Despite advances in integrated management measures of *D. suzukii* [12,13], the prophylactic use of synthetic insecticides (organophosphates, pyrethroids, and spinosyns) is the most adopted tool in the world [14,15]. However, the dependence on synthetic insecticides has been a cause of concern. This has been demonstrated by the negative impact on beneficial insects [16,17,18], the presence of chemical residues in fruits [19], and the selection of resistant populations of *D. suzukii* [14,20]. Continuous applications of insecticides such as spinosad have led to the evolution of pest resistance [15,21]. This scenario is considered to be worrying because it is the only effective product approved for the organic production system to fight *D. suzukii* [15,22].

A possible alternative to mitigate these effects is the use of essential oils (EOs) since they have low toxicity to natural enemies, degrade rapidly, have a reduced environmental impact [23], and are compatible with conventional and organic agriculture [24]. EOs are synthesized by the secondary metabolism of aromatic plants and are a rich source of bioactive molecules that contain terpenes with functional groups such as acids, alcohols, ketones, phenols, hydrocarbons, and others [25]. This composition is important since the toxicity of EOs changes with the number of chemical components and the interactions between them [26,27,28,29]. This also contributes to multiple modes of action, which may be beneficial in preventing or delaying the evolution of pest resistance to insecticides [30].

Some EOs have been tested in the laboratory for the management of *D. suzukii*, including contact insecticides, larvicides, and oviposition and feeding inhibitors [6,24,31]. Some field studies have employed EOs in monitoring bait traps to attract or repel *D. suzukii* adults [32,33]. Linalool, one isolated component of EOs, was shown to cause vapor activity in *D. suzukii* adults [34], whereas thymol showed a significant level of repellency for *D. suzukii* males and females for 24 h [32], and limonene caused changes in the fatty body of the third instar of this insect [6]. In addition, commercial EO-based products such as KeyPlex Ecotrol^®^ PLUS, KeyPlex Sporan^®^ EC^2^ [24], and Naissance Neem Virgin^®^ [35] deterred *D. suzukii* adults. Although sold at reasonable prices [30], it is important to have more in-depth information on the bioactivity of the EOs available on the market and their toxicity to non-target organisms, mainly *Trichopria anastrephae* Lima (Hymenoptera: Diapriidae), to develop effective formulations for *D. suzukii*. *Trichopria anastrephae* is a pupal idiobiont endoparasitoid that is considered to be the natural enemy of strawberries, with the highest occurrence in crops of this plant in Brazil [36]. This parasitoid demonstrated potential for interspecific competition with another pupal parasitoid of *D. suzukii*, *Pachycrepoideus vindemmiae* (Hymenoptera: Pteromalidae) [37]. However, *T. anastrephae* is highly susceptible to the synthetic insecticides used in the management of *D. suzukii* [16,18,38], a fact that may compromise integrated pest management.

In this context, it is necessary to diversify integrated management strategies and reduce the use of synthetic insecticides in the control of *D. suzukii.* Therefore, this study aimed to i) identify and quantify the composition of 19 EOs commonly used and available on the market; ii) evaluate the toxicity of these EOs to *D. suzukii* and *T. anastrephae* adults using ingestion and topical application methods; iii) evaluate the effects of dry residues of EOs on the oviposition behavior of *D. suzukii* in artificial fruits; and iv) verify the repellent effect of EOs in *D. suzukii* females using a dual-choice olfactometer.

## 2. Results

The chromatographic analysis identified and quantified 32 chemical components from the EOs of the 19 studied species, representing 61.3–100.0% of the total sample (Table 1). Hence, chromatographic peaks were classified into hydrocarbon monoterpenes (18.4–51.5%), oxygenated monoterpenes (5.0–96.8%), hydrocarbon sesquiterpenes (9.0–19.1%), oxygenated sesquiterpenes (4.0–7.6%), and arylpropanoids (13.4–86.6%) (Table 1). The chemical profiles revealed that the common components most frequently found were (*E*)-cinnamaldehyde and (*E*)-cinnamyl acetate in *C. verum*; α-pinene in *C. sempervirens*; geranial, neral, and geraniol in *Cymbopogon* spp.; menthol and menthone in *Mentha* spp.; 1,8-cineole, geranial, and geraniol in *Eucalyptus* spp.; terpinen-4-ol in *M. alternifolia*; estragole in *O. basilicum*; and α-bulnesene in *P. cablin*.

Significant mortality of *D. suzukii* adults was found when the toxicity was assessed 120 h after treatment with the 19 EOs of the studied species, both for the topical application (F= 43.12; df = 21, 84; *p* >0.0001) and ingestion (F= 65.44; df = 21, 84; *p* >0.0001) bioassays. In both bioassays, EOs from *C. verum*, *C. martinii*, and *C. winterianus* caused between 75% and 100% mortality in *D. suzukii* adults, the same toxicity as the spinetoram-based insecticide (topical application (98% mortality); ingestion (79% mortality)) (Figure 1). 

From the concentration–response curves and the overlap of the confidence intervals for the topical bioassays, it was found that *C. flexuosus* and the species of the genus *Mentha*, *M. arvensis*, *M. cardiaca*, *M. spicata*, *M. piperita,* and *M. citrata*, had lower CL_50_ and CL_90_ values than other species (Table 2). In contrast, EOs from *C. verum*, *C. citratus* QT *citratus*, and *C. winterianus* showed CL_50_ and CL_90_ values related to the highest toxicities detected in the ingestion bioassays (Table 3).

Regarding the oviposition deterrence, *D. suzukii* females significantly reduced oviposition in artificial fruits (F = 72.10, df = 21, 639, *p* > 0.0001) when exposed to the dry residues of EOs of *C. verum* (2.9 eggs/female) and *C. citratus* QT *citratus* (5.5 eggs/female) and similarly when exposed to the positive control treatment containing spinetoram (3.3 eggs/female) (Table 4). These EOs caused 84.5% (*C. verum*) and 71.3% (*C. citratus* QT *citratus*) oviposition reductions concerning the negative controls water (19.2 eggs/female) and acetone (17.6 eggs/female). In contrast, EOs of other species of the genus *Cymbopogon* spp. (*C. martinii*, *C. flexuosus*, and *C. winterianus*), *Eucalyptus* spp. (*E. radiata* and *E. staigeriana*), and *Mentha* spp. (*M. arvensis*, *M. cardiaca*, *M. spicata*, *M. piperita*, and *M. citrata*) caused intermediate deterrence effects on oviposition (≈ 56.7% reduction) compared to the negative controls (Table 4). EOs of *C. sempervirens*, *C. citratus* QT *myrcene*, *E. globulus*, *E. citriodora*, *M. alternifolia*, *O. basilicum*, and *P. cablin* showed no detrimental effect on the oviposition of females of *D. suzukii* (Table 4). In addition, 90% of *D. suzukii* females showed an avoidance behavior to treatments containing EOs in the repellency bioassay using the dual-choice olfactometer (Figure 2).

Concerning the lethal and sublethal toxicity of EOs from the 19 analyzed species in *T. anastrephae* adults, the observed less than 20% mortality for the topical application (F= 98.10; df= 19, 480; *p* >0.0001) and ingestion (F= 72.34; df= 19, 480; *p* >0.0001) bioassays was statistically similar to the negative controls (Table 5). These treatments differed significantly from the positive controls using the spinetoram-based insecticide, which resulted in a mortality of 29.6% (topical application bioassay) and 35.7% (ingestion bioassay) (Table 5). In addition, all treatments (EOs, negative and positive controls) showed no sublethal effects concerning the parasitism of surviving *T. anastrephae* females over the seven days of evaluation (F = 118.12; df = 19, 378; *p* > 0.0001) (Table 5).

## 3. Discussion

This study presents evidence of the lethal and sublethal effects of the 19 analyzed and commercially available EOs on populations of *D. suzukii* and *T. anastrephae*. The chemical profile of these EOs was determined qualitatively and quantitatively by applying the GC-MS technique. Our phytochemical analyses revealed that the most frequent compounds found in these EOs were linalool (5.0 and 70.8%), neral (9.1 and 28.7%), geraniol (5.5 and 91.7%), geranial (22.1 and 44.6%), menthol (45.3 and 96.8%), and 1,8-cineole (36.8 and 40.1%). Although the chemical composition differs among the 19 selected EOs, there are reports of common functional groups of monoterpenes, such as α-pinene, citral, citronellal, citronellol, linalool, menthol, menthol, and 1,8-cineol, that have multiple chemical properties against *D. suzukii* such as toxicity, repellency, and deterrence of feed and oviposition [6,31,34,39]. It is also known that the efficacy of EOs on *D. suzukii* depends on the composition and proportions of the main substances, as demonstrated for *C. verum* [31] and *Gaultheria fragrantima* [40]. Likewise, the interactions of minority constituents contained in EOs have been reported to have synergistic action, providing a significant increase in the effectiveness of formulations for arthropod pests [26].

Our study showed that the EOs of *C. verum, C. martinii,* and *C. winterianus* were toxic in *D. suzukii* adults subjected to topical application and ingestion bioassays. This information supports the hypothesis that all components in the EOs show a significant positive correlation with the mortality of flies, either by their combination or an adequate proportion of these substances [27,28,29]. Accordingly, studies by [31] observed that *D. suzukii* is susceptible to the toxic action of EOs, particularly from *C. verum* and *Cymbopogon* spp. Moreover, other studies have shown that EOs of the genera *Cinnamomum* and *Cymbopogon* are toxic to Diptera [41,42].

Regarding the toxicity related to topical application, the EOs of *C. flexuosus* and *Mentha* spp. (*M. arvensis*, *M. cardiaca*, *M. spicata*, *M. piperita*, and *M. citrata*) showed the lowest concentration–response 120 h after treatments, with LC_50_ and LC_90_ of 5.07 and 17.89 mg/L, respectively. In contrast, the LC_50_ and LC_90_ values in the ingestion bioassay were 15.13 and 35.67 mg/L, respectively. Thus, the ingestion bioassay required higher EO concentrations than the topical application bioassay to induce toxicity. A differential efficacy in inducing mortality has also been reported between topical application and ingestion bioassays in *D. suzukii* for various essential oils [22,31,43]. This fact may be related to the differential penetration of substances in the insects’ hemolymph and the metabolizing and excretion time required for each treatment [44]. As EOs are complex mixtures of compounds, they may act on more than one site. The rapid action of EOs indicates that terpenes passing through the insect respiratory system interfere with physiological functions [6,45] or act as acetylcholinesterase inhibitors, interacting with the neurotransmitter octopamine and the gamma-aminobutyric acid receptor [30,46]. In this study, we observed that the EOs increased the impulsive activity of the *D. suzukii* nervous system, resulting in spasms, impaired coordination, agitation, and shivering. This observation agrees with other studies on the effects of EOs on the central nervous system reporting excessive stimulation of the motor nerves causing paralysis and death [39,40,47].

In addition to its lethal toxicity, EOs can alter the oviposition behavior of *D. suzukii*, causing a reduction in the number of eggs per fruit or the suppression of oviposition activity, and affect the sense of orientation of the insects [6,31,32,43]. Under a no-choice scenario, the EOs of *C. verum* and *C. citratus* QT *citratus* had deterrent effects on oviposition, reducing the number of eggs laid per fruit by 84.5 and 71.3%, respectively. Coincidently, studies by [31] observed that *D. suzukii* is susceptible to the toxic action of the EOs of *C. verum* and *C. citratus* and their major components cinnamaldehyde and citral (geranial+neral). Moreover, a *C. verum*-based extract (Progranic Gamma, PLM^®^) significantly reduced the number of *D. suzukii* larvae in treated raspberries in field experiments [48]. In light of the high reproductive potential of *D. suzukii*, with an average of 1649 eggs per female life span [49], compounds that interfere with this reproductive performance can be useful in reducing the population density of the species. The oviposition deterrence is a critical aspect considering that the biggest damage caused by *D. suzukii* infestation is due to the rupture of the fruit epidermis for egg deposition, mainly in fruits with a thin epidermis [22,36,50]. Thus, the use of EOs with a dissuasive effect on oviposition may decrease the degree of rupture of the fruit epidermis and, consequently, significantly reduce the infestation with pathogens that accelerate the fruit deterioration [22,31,43]. In addition, the observation of 90% of *D. suzukii* females displaying an avoidance behavior when exposed to EOs in the dual-choice olfactometry tests confirms that these substances guide flies in search of suitable hosts [33,51].

In addition to the toxicity of the studied EOs on *D. suzukii*, it is also necessary to consider the natural enemies used to manage these pest populations [17,31,43]. According to our results, all analyzed EOs caused low mortality in *T. anastrephae* adults through topical application and ingestion bioassays. This fact is of fundamental importance for the management of *D. suzukii* in the field since *T. anastrephae* is the most promising endemic parasitoid for the biological control of *D. suzukii* in Brazil [36,37,38]. Accordingly, studies [43,44] have reported that the EOs of *Piper* spp. (Piperaceae) and *Rosmarinus officinalis* (Lamiaceae) had low side effects on *T. anastrephae* and high toxicity on *D. suzukii*. The management of *D. suzukii* by control strategies using EOs could be more compatible with biological control methods than conventional insecticides, resulting in safe programs for integrated pest management [16,17,18].

Hence, the studied EOs from *C. verum*, *Cymbopognom* spp., and *Mentha* spp., which differ in their topical toxicity and deterrence on food absorption and oviposition, are the most promising alternatives among the analyzed EOs to be included in the integrated management programs of *D. suzukii* since they have insecticidal properties and selectivity to non-target organisms such as *T. anastrephae*. These EOs can also be used in organic farming systems where synthetic substances are forbidden for pest management. However, the stability of EOs in the field and the persistence of their effects over time are often described as limited [30]. Thus, from an economic point of view, formulations of EOs, by emulsion or encapsulation, are needed to allow EOs to increase their biological activity and stability [52,53]. These concerns deserve attention and further research for product development directed to sustainable agriculture.

## 4. Materials and Methods

### 4.1. Insect Breeding

*Drosophila suzukii* were reared in glass tubes (5 mL in volume; 7.5 cm in height × 1.2 cm in diameter) containing an artificial diet based on corn flour, yeast, and sugar and buffered with hydrophilic cotton [49]. Adults were transferred to new tubes with fresh food twice a week. The laboratory population was established in January 2018 from the collection of fruits from organic strawberries infested with larvae and grown in an experimental greenhouse in Curitiba, Paraná, Brazil (31° 38’ 20’’ S, 52° 30’ 43’’ W).

The population of *T. anastrephae* was formed from organic mulberry fruits infested by *D. suzukii* in Pelotas, Rio Grande do Sul, Brazil (31°38′ 20″ S, 52° 30′ 43″ W) in 2018. In the laboratory, *T. anastrephae* parasitoids were reared and multiplied in *D. suzukii* pupae, as suggested by [54]. *Trichopria*
*anastrephae* adults were fed with an 80% (p:v) (honey: water solution).

Both *D. suzukii* and *T. anastrephae* are colonies susceptible to chemical insecticides, being maintained under controlled conditions (25 ± 2 °C, 70 ± 10% relative humidity, and 12:12 h photophase (light: dark). Specimens from the organic greenhouse were introduced annually to the laboratory populations to maintain their genetic variability.

### 4.2. Essential Oils: Source and GC-MS Analysis

Table 1 details the origin of the 19 EOs samples used in this study. Industrial-scale extractions by dragging water vapor were applied to obtain the EOs, which were selected based on their commercial availability. All samples were standardized according to the methods and quality indicators described in the *Brazilian Pharmacopoeia* (≥ 90% purity).

The chemical composition of the EOs was assessed by gas chromatography/mass spectrum (GC/MS) performed with a Shimadzu 2030 equipment coupled to a sequential Shimadzu TQ8040 mass detector, using an HP-5MS column (30 m × 0.25 mm × 0.25 μm). Analytical conditions were established at 250 and 260 °C injector and transfer line temperatures, respectively; the oven temperature was programmed from 60 to 240 °C, at a rate of 3 °C min^−1^ and maintained at 240 °C for 10 min; helium gas at 1.0 mL.min^−1^; 0.1 μL injection volume (5% HPLC grade n-hexane solution); and a 1:30 split ratio.

The identification of the components was performed by comparing the mass spectra with those of commercial libraries [55], and by their linear retention rates [56] after injection of a homologous series of alkanes (C_8_–C_26_) under the same experimental conditions, and compared with data from the literature [57].

### 4.3. Bioassays

Seven-day-old adults of *D. suzukii* and *T. anastrephae* were deprived of food (artificial diet and honey, respectively) for 8 h, with a regular water supply. The 19 EOs were diluted in acetone (PanReac-UV-IR-HPLC-GPC PAI-ACS, 99.9%) to obtain solutions at 2.5, 5.0, 7.5, 10, 20, 40, and 80 mg L^−1^. A synthetic insecticide based on spinetoram (Delegate 250 WG™) was used as a positive control for the field dosage at 250 g of active ingredient per L of water. Water and acetone were used as negative controls. All bioassays were performed in the laboratory under controlled conditions at 25 ± 2 °C, 70 ± 10% relative humidity, and 12:12 h light:dark photoperiod.

#### 4.3.1. Essential Oil Toxicity on *D. suzukii*

The methods of ingestion and topical application were used to assess the toxicity of essential oil on *D. suzukii*. For the bioassay of topical application, 10 female and 10 male adults of *D. suzukii* from the maintenance breeding were kept for 60 s in transparent glass tubes (V = 5 mL, 7.5 cm height × 1.2 cm diameter), with the top sealed with cotton moistened with ethyl ether. Then, the sedated insects were transferred to a 9-cm-diameter Petri dish (sampling unit) and, using a Potter’s Tower (Burkard Scientific Uxbridge, United Kingdom), 2 mL of the EOs was applied per sampling unit. Afterward, these units were packed in transparent plastic cages (V = 700 mL, 4.7 cm height × 6.7 cm diameter) closed with a voile mesh, allowing ventilation to air exchange between the inner and outer environment of the cage. *D. suzukii* adults were fed with water honey solution until the end of the evaluations, as proposed by [54].

For the ingestion bioassay, eight female and eight male *D. suzukii* adults from the maintenance breeding were kept in transparent plastic cages (V = 700 mL, 4.7 cm height × 6.7 cm diameter) sealed with a voile mesh. Subsequently, hydrophilic cotton rovings saturated with EO solutions stored in glass tubes (V = 10 mL, 4.7 cm height × 6.7 cm diameter) were added to these cages and kept for 24 h. After the exposure period, the tubes containing the cotton with OES were replaced by distilled water and honey until the end of the evaluation period.

The experimental design was completely randomized for both bioassays. For each EO, 7 concentrations (2.5, 5.0, 7.5, 10, 20, 40, and 80 mg L^−1^) were assessed, establishing 4 replicates (cages) with 20 adults for the topical bioassay (*n* = 560) and 5 replicates with 16 adults in the ingestion bioassay (*n* = 560). Mortality was assessed at intervals of 24 h after the exposure treatments until 120 h after the treatment. Insects that did not react to the touch of a fine tip brush were considered dead. Corrected mortality was calculated using Abbott’s formula [58].

#### 4.3.2. Concentration–Response Curves

Based on the results of the toxicity bioassay, the treatments and positive control (spinetoram) were evaluated to estimate the concentration required to kill 50 and 90% of the exposed flies (lethal concentration (CL); CL_50_ and CL_90_, respectively). For this purpose, seven concentration values were tested for each treatment (from 2.5–80 mg L^−1^ for EOs and 5–75 mg L^−1^ for spinetoram), and the exposure mode was based on Finney (1971). The procedures and criteria for exposure and evaluation were identical to the toxicity bioassays. In the topical application bioassays, 4 replicates (cages) with 20 flies (*n* = 80) were used for each concentration of each product (*n* = 560). While in the ingestion bioassays, 5 replicates (cages) with 16 flies (*n* = 80) were used for each concentration of each product (*n* = 560).

#### 4.3.3. Oviposition Bioassay

Artificial fruits were used as a substrate to evaluate the deterrent effect of EOs on the oviposition of *D. suzukii.* These fruits were made from agar (19 g), raspberry gelatin (10 g), methylparaben (0.8 g dissolved in 8 mL of 99.9% ethyl alcohol), and 850 mL of distilled water. Then, each fruit was individually immersed for 60 s in 4 mL of one of the 19 EO solutions prepared at a maximum concentration of 80 mg L^−1^. Afterward, the fruits were conditioned at 25 ± 2 °C and 70% ± 10% humidity for 4 h on filter paper to evaporate the excess moisture and for the deposition of residues from the EOs. The fruits obtained by these means were individually placed inside 500 mL plastic cages, inside of which 5 female and 5 male 7-day-old *D. suzukii* flies were released. After 24 h, the insects were removed, and the eggs in each artificial fruit were counted under a stereo microscope with 40X magnification. The experimental design of this bioassay, consisting of 30 replicates (fruits) per treatment, was entirely randomized.

#### 4.3.4. Repellence Bioassay

*D. suzukii* females up to 24 h old were used for the repellency test. Each female was placed alone inside a glass tube (1.3 cm diameter × 10 cm length) to verify the repellent effect of the EOs, using acetone as the negative control. Subsequently, the glass tube containing the female was connected to a dual-choice glass olfactometer with a diameter of 8.0 cm and an initial compartment of 20 cm on each side and placed under fluorescent light (60 W, 290 lx) as previously suggested by [6]. A filter paper (4.0 cm width × 10.0 cm length) folded in the shape of an accordion and containing 5 µL of the analyzed EO (Table 1) was added at the end of one of the olfactometer arms. Another filter paper of the same size and containing 5 µL of acetone (negative control) was added to the other arm of the olfactometer. Afterward, airflow was provided into the system at a rate of 0.8 L min^− 1^ from an air source previously filtered with active carbon and humidified in distilled water. The olfactometer was washed with neutral soap and hexane and dried in a sterilization oven at 150 °C every 4 replicates (4 tested females). After this cleaning process, the substances (EOs or acetone) were replaced, and the test continued. For each analyzed EO, 40 replicates (one female per replicate, *n* = 40) were conducted. The responses were considered positive when *D. suzukii* females reached the odor source (EOs or acetone), walked at least 10 cm within the olfactometer arms, and remained at this position for at least 1 min. Females that did not move toward any of the 2 olfactory sources until 1 min after connecting the glass tube with the olfactometer were discarded from the test.

#### 4.3.5. Essential Oil Toxicity in *T. anastrephae*

The most active treatments for *D. suzukii* in previous toxicity bioassays were used to assess toxicity in *T. anastrephae* adults. From this, the LC_90_ values of the EOs from the 120 h post-exposure evaluation, determined in the concentration–response curve bioassays, were used. The toxicity in *T. anastrephae* adults was verified by ingestion and topical application bioassays in a Potter tower as previously described for tests on *D. suzukii*. Parasitoid mortality was assessed 120 h after the beginning of treatments. The experimental design was completely randomized, with 10 replicates per treatment.

The sublethal effects of the ingestion and topic bioassays treatments in the surviving insects were assessed by 7 days of daily exposure of 24-h-old *D. suzukii* pupae to each of the 5-days-old female *T. anastrephae* survivors, following the methodology proposed by [54]. During the experiment period, surviving *T. anastrephae* females were fed with 80% honey. After 24 h of exposure, the pupae offered to the *T. anastrephae* females were removed and packed into 100 mL plastic cups sealed with voile until the emergence of the insects (fly or parasitoid). Percent parasitism was determined considering the total number of parasitoids concerning the number of offered pupae.

### 4.4. Data Analyses

The studied variables were analyzed using the generalized linear models of the exponential family of distributions [59]. The comparison between treatments (essential oils and positive or negative control) was determined using a one-way ANOVA, and the difference between each treatment was compared using a Dunnett’s post hoc test, using the “R” statistical software version 2.15.1 [60]. A binomial model with a complementary log-log link function (gompit model) was used to estimate the lethal concentrations (LC_50_ and LC_90_), implementing the *Probit Procedure* in the SAS software version 9.2 [61]. Finally, the repellency percentage (RP) was calculated for each EO according to [62], using the formula PR (%) = [(Nc − Nt)/(Nc + Nt)] × 100, where Nc is the number of insects in half of the negative control group, and Nt is the number of insects in half of the treated group.

## 5. Conclusions

Chromatographic analysis of the 19 EO samples used in this study revealed that the common components most frequently found were (*E*)-cinnamaldehyde and (*E*)-cinnamyl acetate in *C. verum*; α-pinene in *C. sempervirens*; geranial, neral, and geraniol in *Cymbopogon* spp.; menthol and menthone in *Mentha* spp.; 1,8-cineole, geranial, and geraniol in *Eucalyptus* spp.; terpinen-4-ol in *M. alternifolia*; estragole in *O. basilicum*; and α-bulnesene in *P. cablin*. In particular, this is the first report of the use of essential oil of *C. martinii*, *C. flexuosus*, *E. staigeriana, M. cardiaca, M. spicata*, and *M. citrata* for the control of *D. suzukii*, and the first record for *T. anastrephae*. Herein, EOs of *C. verum*, *C. citratus* QT *citratus*, and *C. winterianus* showed the highest toxicity in the ingestion bioassay for *D. suzukii*. The dry residues of *C. verum* and *C. citratus* QT *citratus* reduced the oviposition of *D. suzukii* and the 19 EOs analyzed repelled *D. suzukii* females. Interestingly, these EOs provided low mortality for *T. anastrephae* and did not cause a reduction in the parasitism of surviving *T. anastrephae* females.

## Figures and Tables

**Figure 1 molecules-27-06215-f001:**
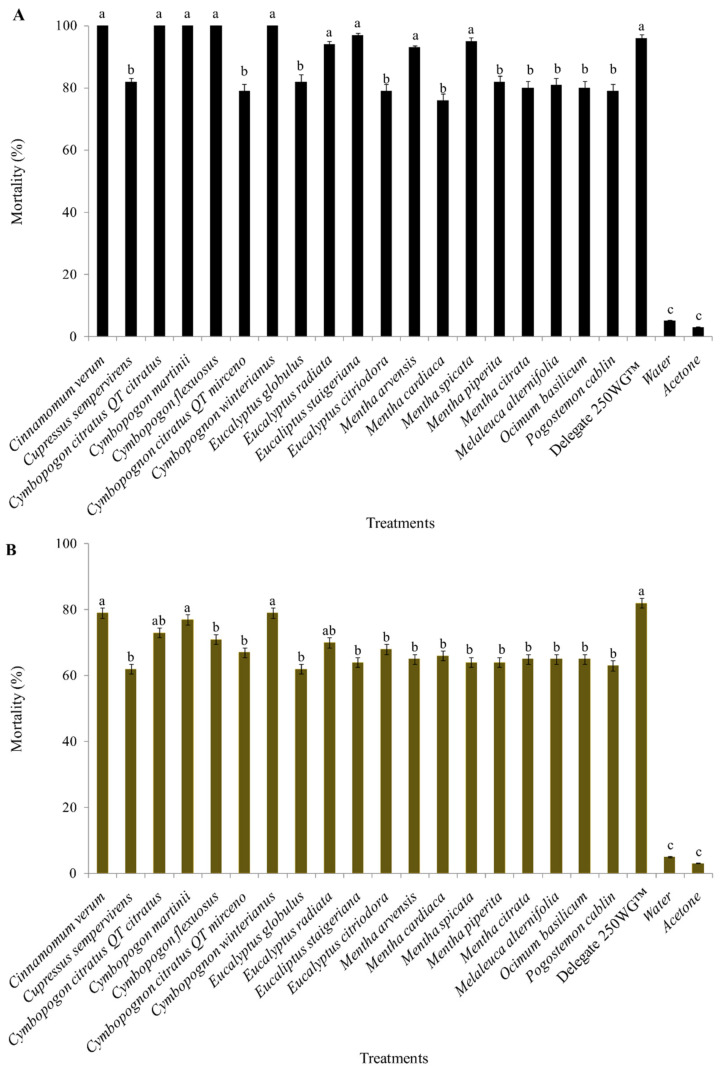
Mortality of *Drosophila suzukii* when treated with essential oils from different plant species in a topical application (**A**) and ingestion bioassays (**B**). Data are presented as the mean ± standard error. Means followed by followed by different letters (a, ab, b or c) in the columns of each figure indicate significant differences between treatments (one-way ANOVA, Dunnett test, *p* < 0.05).

**Figure 2 molecules-27-06215-f002:**
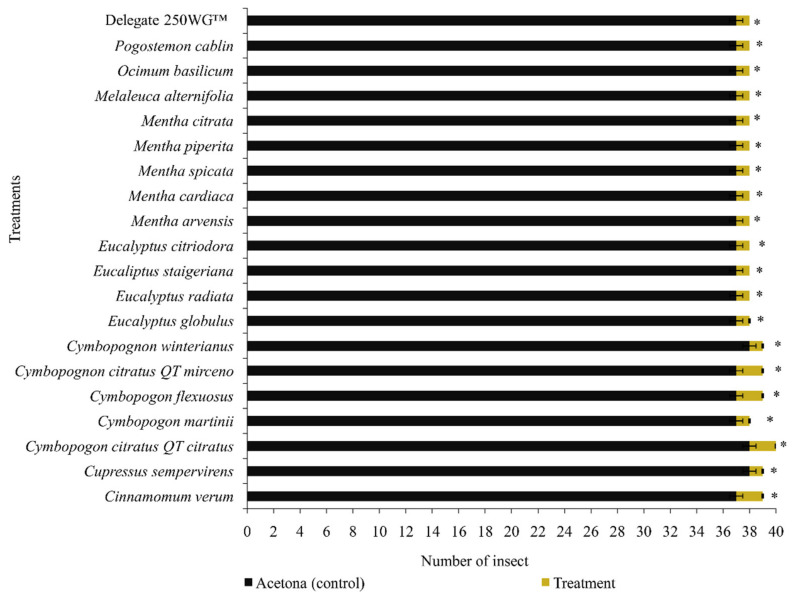
Repellency test bioassay with *Drosophila suzukii* adults inside a dual-choice olfactometer. Data are presented as the mean ± standard error. * Values are means obtained after 40 replicates. The mean numbers of adults were compared using paired *t*-tests at a significance level of *p* ≤ 0.05. Asterisks indicate a significant difference between controls and treatments.

**Table 1 molecules-27-06215-t001:** Essential oils evaluated for the management of *Drosophila suzukii* and *Trichopria anastrephae*.

Constituents	RI *	% Peak Area																		
		1 **	2	3	4	5	6	7	8	9	10	11	12	13	14	15	16	17	18	19
α-pinene	942	---	51.5	---	---	---	---	---	---	---	---	---	---	---	---	---	---	---	---	---
α-terpinene	1016	---	---	---	---	---	---	---	---	---	---	---	---	---	---	---	---	18.4	---	---
3-carene	1022	---	28.1	---	---	---	---	---	---	---	---	---	---	---	---	---	---	---	---	---
γ-terpinene	1054	---	---	---	---	---	---	---	---	---	---	---	---	---	---	---	---	23.3	---	---
Hydrocarbon monoterpene		0	79.6	0	0	0	0	0	0	0	0	0	0	0	0	0	0	41.7	0	0
1,8-cineole	1030	---	---	---	---	---	---	---	---	---	36.8	---	8.4	---	40.1	---	---	---	---	---
linalool	1096	---	---	5.0	10.6	10.3	11.0	10.3	---	6.9	---	---	---	---	10.3	---	---	---	23.2	---
cis-2-p-menthen-1-ol	1116	---	---	---	11.0	---	---	---	---	---	---	---	---	---	---	---	---	---	---	---
trans-2-p-menthen-1-ol	1136	---	---	---	10.2	---	---	---	---	---	---	---	---	---	---	---	---	---	---	---
menthone	1148	---	---	---	---	---	---	---	---	---	---	---	---	---	14.5	16.0	---	---	---	---
citronellal	1151	---	---	---	---	---	---	68.1	---	---	---	40.2	---	---	---	---	---	---	---	---
iso-menthone	1158	---	---	---	---	---	---	---	---	---	---	---	---	---	---	---	18.8	---	---	---
menthol	1177	---	---	---	---	---	---	---	---	---	---	---	---	96.8	---	45.3	56.4	---	---	---
terpinen-4-ol	1180	---	---	---	---	---	---	---	---	---	---	---	---	---	---	---	---	49.9	---	---
α-terpineol	1186	---	---	---	13.8	---	---	---	86.8	---	---	---	---	---	---	---	---	8.3	---	---
citronellol	1222	---	---	---	---	---	---	21.6	---	---	---	16.7	---	---	---	---	---	---	---	---
neral	1233	---	---	17.6	---	28.7	19.4	---	---	12.8	25.8	---	---	---	---	---	---	---	---	---
carvone	1239	---	---	---	---	---	---	---	---	---	---	---	---	---	---	---	20.0	---	---	---
geraniol	1246	---	---	18.9	---	8.9	5.5	---	13.2	17.6	---	28.6	91.6	---	---	---	---	---	---	---
piperitone	1253	---	---	---	54.3	---	---	---	---	---	---	---	---	---	---	---	---	---	---	---
geranial	1261	---	---	39.5	---	44.6	60.5	---	---	24.1	22.1	---	---	---	---	---	---	---	---	---
Oxygenated monoterpene		0	0	81.0	99.9	85.0	96.4	100	100	61.4	84.7	85.5	100	96.8	64.9	61.3	95.2	58.2	23.2	0
β-patchoulene	1385	---	---	---	---	---	---	---	---	---	---	---	---	---	---	---	---	---	---	9.0
α-guaiene	1437	---	---	---	---	---	---	---	---	---	---	---	---	---	---	---	---	---	---	19.0
seychellene	1450	---	---	---	---	---	---	---	---	---	---	---	---	---	---	---	---	---	---	19.1
α-patchoulene	1460	---	---	---	---	---	---	---	---	---	---	---	---	---	---	---	---	---	---	16.2
α-bulnesene	1504	---	---	---	---	---	---	---	---	---	---	---	---	---	---	---	---	---	---	15.1
Hydrocarbon sesquiterpene		0	0	0	0	0	0	0	0	0	0	0	0	0	0	0	0	0	0	78.4
cedrol	1608	---	7.6	---	---	---	---	---	---	---	---	---	---	---	---	---	---	---	---	---
bulnesol	1668	---	---	---	---	---	---	---	---	---	---	---	---	---	---	---	---	---	---	4.0
Oxygenated sesquiterpene		0	7.6	0	0	0	0	0	0	0	0	0	0	0	0	0	0	0	0	4.0
estragole	1197	---	---	---	---	---	---	---	---	---	---	---	---	---	---	---	---	---	76.7	---
(*E*)-cynnamaldeyde	1270	86.6	---	---	---	---	---	---	---	---	---	---	---	---	---	---	---	---	---	---
(*E*)-cinnamyl acetate	1441	13.4	---	---	---	---	---	---	---	---	---	---	---	---	---	---	---	---	---	---
Arylpropanoid		100.0	0	0	0	0	0	0	0	0	0	0	0	0	0	0	0	0	76.7	0
methyl nerolate	1290	---	---	---	---	---	---	---	---	14.2	---	---	---	---	---	---	---	---	---	---
nerol acetate	1362	---	---	19.0	---	7.5	---	---	---	19.1	---	---	---	---	---	---	---	---	---	---
Ester		0	0	19.0	---	7.5	0	0	0	0	0	0	0	0	0	0	0	0	0	0
Total of identification (%)		100	87.2	100	99.9	92.5	96.4	100	100	94.7	84.7	85.5	100	96.8	64.9	61.3	95.2	99.9	99.9	82.4

* RI = Calculated Retention Index. ** Species: 1. *Cinnamomum verum* (cinnamon), 2. *Cupressus sempervirens* (cypress), 3. *Cymbopogon citratus QT citratus* (lemongrass), 4. *Cymbopogon martini* (palmarosa), 5. *Cymbopogon flexuosus* (lemongrass), 6. *Cymbopognon citratus QT myrcene* (lemongrass), 7. *Cymbopognon winterianus* (citronella grass), 8. *Eucalyptus globulus* (blue gum), 9. *Eucalyptus radiata* (Forth River peppermint), 10. *Eucaliptus staigeriana* (lemon Ironbark), 11. *Eucalyptus citriodora* (citriodora), 12. *Mentha arvensis* (wildmint), 13. *Mentha cardiaca* (gingermint), 14. *Mentha spicata* (spearmint), 15. *Mentha piperita* (peppermint), 16. *Mentha citrata* (eau de cologne mint), 17. *Melaleuca alternifolia* (tea tree), 18. *Ocimum basilicum* (basil) and 19. *Pogostemon cablin* (patchouli). Origin/manufacturer of species: Phytoterapica Industrial Ltd.a. ^1^, Dhonella Industrial Ltd.a. ^2^, Laszlo Industrial Ltd.a. ^3,4,5,6,13,18,19^, BioEssência Industrial Ltd.a. ^7,15^, Oshadhi Industrial Ltd.a. ^8,9,11^, Terra Flor Industrial Ltd.a. ^10,12,16,17^, Now Food Industrial Ltd.a. ^14^ --- Constituents not present.

**Table 2 molecules-27-06215-t002:** Estimates of the LC_50_ and LC_90_ values (mg L^−1^) and confidence interval calculated 120 h after bioassays of the topical application of essential oils and the spinetoram-based synthetic insecticide (Delegate 250 WG™) on *Drosophila suzukii* adults.

Treatments	Slope ± SE	LC_50_ (95% CI) ^a^	LC_90_ (95% CI) ^b^	χ2 ^c^	df ^d^
*Cinnamomum verum,*	2.74 ± 0.21	11.02 (10.12–13.45)	17.12 (16.10–18.11)	5.44	6
*Cupressus sempervirens*	2.78 ± 0.17	14.45 (13.76–15.44)	24.21 (22.13–25.11)	7.11	6
*Cymbopogon citratus QT citratus*	3.11 ± 0.11	10.12 (8.11–12.78)	16.18 (15.11–17.98)	8.12	6
*Cymbopogon martinii*	2.98 ± 0.21	11.73 (10.44–13.79)	17.10 (16.13–18.11)	5.10	6
*Cymbopogon flexuosus*	3.07 ± 0.16	6.11 (5.75–7.43)	17.68 (16.13–19.17)	6.03	6
*Cymbopognon citratus QT myrcene*	2.75 ± 0.11	12.67 (11.74–14.90)	26.12 (25.01–28.17)	5.44	6
*Cymbopognon winterianus*	2.74 ± 0.22	11.54 (10.23–14.98)	18.17 (16.08–19.10)	9.78	6
*Eucalyptus globulus*	2.95 ± 0.11	14.10 (12.76–16.89)	24.56 (22.13–27.89)	8.12	6
*Eucalyptus radiata*	2.87 ± 0.14	10.43 (9.72–11.13)	18.15 (16.19–20.11)	6.13	6
*Eucalyptus staigeriana*	3.08 ± 0.12	11.23 (9.55–13.75)	16.12 (15.97–17.01)	7.11	6
*Eucalyptus citriodora*	2.97 ± 0.22	12.12 (11.75–14.20)	23.44 (21.76–25.16)	8.19.	6
*Mentha arvensis*	3.10 ± 0.14	5.07 (3.11–6.10)	15.62 (14.45–17.12)	9.90	6
*Mentha cardiaca*	2.98 ± 0.16	7.10 (5.95–9.24)	16.23 (14.54–18.23)	8.16	6
*Mentha spicata*	3.08 ± 0.18	8.14 (6.04–9.74)	17.34 (15.23–18.12)	7.35	6
*Mentha piperita*	2.95 ± 0.13	7.98 (5.24–9.11)	16.78 (15.89–19.11)	7.14	6
*Mentha citrata*	2.67 ± 0.15	8.97 (6.07–11.24)	17.89 (16.78–19.34)	8.45	6
*Melaleuca alternifolia*	2.87 ± 0.10	17.13 (15-60–19.20)	22.78 (21.15–24.97)	8.07	6
*Ocimum basilicum*	3.07 ± 0.14	18.10 (17.74–20.05)	23.44 (22.14–25.17)	6.40	6
*Pogostemon cablin*	2.76 ± 0.16	19.28 (17.18–22.40)	20.79 (19.11–23.44)	5.53	6
Delegate 250 WG™	2.78 ± 0.09	30.12 (28.75–32.44)	26.23 (24.24–28.79	6.12	6

^a^ LC_50_ and ^b^ LC_90_: Insecticide concentrations (mg L^−1^) required to kill 50 or 90% of the adults of *D. suzukii*, respectively (CI: confidence interval at 95% error probability); ^c^ χ^2^: Pearson’s chi-square value; ^d^ df: degrees of freedom.

**Table 3 molecules-27-06215-t003:** Estimates of the LC_50_ and LC_90_ values (mg L^−1^) and confidence interval calculated 120 h after bioassays of the ingestion of essential oils and the spinetoram-based synthetic insecticide (Delegate 250 WG™) on *Drosophila suzukii* adults.

Treatments	Slope ± SE	LC_50_ (95% CI) ^a^	LC_90_ (95% CI) ^b^	χ2 ^c^	df ^d^
*Cinnamomum verum,*	2.74 ± 0.21	15.78 (12.10–17.14)	25.67 (23.44–28.97)	7.10	6
*Cupressus sempervirens*	2.78 ± 0.17	15.16 (13.14–18.10)	32.16 (30.11–35.78)	9.11	6
*Cymbopogon citratus QT citratus*	3.11 ± 0.11	14.11 (13.14–18.67)	27.11 (25.16–29.98)	8.13	6
*Cymbopogon martinii*	2.98 ± 0.21	16.18 (14.76–18.45)	28.78 (26.17–29.40)	5.44	6
*Cymbopogon flexuosus*	3.07 ± 0.16	16.54 (15.11–19.76)	32.97 (30.11–35.67)	6.78	6
*Cymbopognon citratus QT myrcene*	2.75 ± 0.11	17.86 (16.10–20.06)	33.67 (31.98–35.40)	6.56	6
*Cymbopognon winterianus*	2.74 ± 0.22	15.67 (14.15–19.73)	25.44 (24.56–28.76)	7.10	6
*Eucalyptus globulus*	2.95 ± 0.11	18.34 (17.54–20.13)	31.45 (30.08–33.45)	8.12	6
*Eucalyptus radiata*	2.87 ± 0.14	16.14 (13.74–18.89)	27.65 (26.11–29.80)	9.11	6
*Eucalyptus staigeriana*	3.08 ± 0.12	15.17 (15.10–18.56)	30.24 (29.78–33.70)	6.14	6
*Eucalyptus citriodora*	2.97 ± 0.22	19.16 (17.67–20.14)	32.45 (31.90–35.76)	8.14	6
*Mentha arvensis*	3.10 ± 0.14	17.98 (15.11–21.34)	34.56 (33.78–37.89)	7.70	6
*Mentha cardiaca*	2.98 ± 0.16	16.74 (14.72–19.24)	33.67 (32.89–38.75)	8.15	6
*Mentha spicata*	3.08 ± 0.18	15.13 (13.20–18.19)	34.89 (33.45–37.90)	8.19.	6
*Mentha piperia*	2.95 ± 0.13	20.11 (17.18–22.36)	35.67 (33.14–38.97)	8.23	6
*Mentha citrata*	2.67 ± 0.15	18.10 (16.34–19.55)	34.65 (33.98–37.80)	8.01	6
*Melaleuca alternifolia*	2.87 ± 0.10	17.12 (15.78–21.54)	35.67 (34.98–38.11)	9.70	6
*Ocimum basilicum*	3.07 ± 0.14	17.54 (14.98–18.76)	34.78 (32.45–38.12)	9.08	6
*Pogostemon cablin*	2.76 ± 0.16	18.34 (16.76–22.17)	36.78 (34.50–39.11)	6.23	6
Delegate 250 WG™	2.78 ± 0.09	25.67 (22.34–28.45)	82.34 (80.11–85.67)	7.79	6

^a^ LC_50_ and ^b^ LC_90_: Insecticide concentrations (mg L^−1^) required to kill 50 or 90% of the *D. suzukii* adults, respectively (CI: confidence interval at 95% error probability); ^c^ χ^2^: Pearson’s chi-square value; ^d^ df: degrees of freedom.

**Table 4 molecules-27-06215-t004:** Effect of essential oils on the oviposition behavior of *Drosophila suzukii*.

Treatments	Number of Eggs ^a^	Reduction of Oviposition (%)
*Cinnamomum verum*	2.9 ± 0.32 C	84.5
*Cupressus sempervirens*	17.1 ± 0.43 A	10.9
*Cymbopogon citratus QT citratus*	5.5 ± 0.39 BC	71.3%
*Cymbopogon martinii*	8.8 ± 0.59 B	54.2
*Cymbopogon flexuosus*	7.9 ± 0.58 B	58.8
*Cymbopognon citratus QT myrcene*	17.3 ± 0.97 A	9.9
*Cymbopognon winterianus*	7.2 ± 0.32 B	62.5
*Eucalyptus globulus*	18.1 ± 0.67 A	5.7
*Eucalyptus radiata*	7.3 ± 0.38 B	61.9
*Eucalyptus staigeriana*	7.9 ± 0.87 B	58.8
*Eucalyptus citriodora*	16.9 ±0.98 A	11.9
*Mentha arvensis*	10.0 ± 0.62 B	47.9
*Mentha cardiaca*	7.8 ± 0.55 B	59.3
*Mentha spicata*	10.6 ± 0.62 B	44.7
*Mentha piperita*	7.9 ± 0.58 B	58.8
*Mentha citrata*	7.6 ± 0.37 B	60.4
*Melaleuca alternifolia*	17.6 ± 1.12 A	8.3
*Ocimum basilicum*	18.1 ± 1.75 A	5.7
*Pogostemon cablin*	18.4 ± 0.52 A	4.1
Delegate 250 WG™	3.3 ± 0.55 C	82.8
F	64.12	
df	21, 639	
*p*	> 0.0001	

^a^ Columns with the same letter (A, B, BC or C) are not significantly different (one-way ANOVA, Dunnett test, *p* < 0.05). Data are presented as the mean ± standard error. Each experiment was performed 3 times with 30 randomly selected artificial fruits.

**Table 5 molecules-27-06215-t005:** Mortality and parasitism of *Trichopria anastrephae* at 120 h after exposure to treatments in ingestion and topical application bioassays.

Treatments	Mortality ^a^		P (%) ^a^
	Topical Application	Ingestion	
*Cinnamomum verum*	7.1 ± 0.23 A	19.1 ± 1.15 A	50.1 ± 2.75 A
*Cupressus sempervirens*	8.5 ± 0.11 A	10.2 ± 0.88 A	48.9 ± 1.89 A
*Cymbopogon citratus QT citratus*	4.2 ± 0.15 A	14.3 ± 1.10 A	44.3 ± 1.12 A
*Cymbopogon martinii*	6.4 ± 0.11 A	12.2 ± 1.78 A	50.4 ± 2.10 A
*Cymbopogon flexuosus*	5.5 ± 0.78 A	12.3 ± 1.15 A	49.2 ± 3.07 A
*Cymbopognon citratus QT myrcene*	6.1 ± 0.45 A	10.3 ± 0.77 A	46.1 ± 2.15 A
*Cymbopognon winterianus*	4.5 ± 0.21 A	16.1 ± 1.11 A	45.2 ± 2.10 A
*Eucalyptus globulus*	5.2 ± 0.56 A	15.4 ± 1.34 A	47.3 ± 1.75 A
*Eucalyptus radiata*	6.4 ± 0.67 A	18.4 ± 1.02 A	49.9 ± 2.01 A
*Eucalyptus staigeriana*	7.3 ± 0.34 A	16.7 ± 1.32 A	49.3 ± 1.86 A
*Eucalyptus citriodora*	6.8 ± 0.24 A	15.6 ± 1.44 A	50.2 ± 1.67 A
*Mentha arvensis*	6.7 ± 0.33 A	13.2 ± 1.09 A	43.4 ± 1.65 A
*Mentha cardiaca*	5.7 ± 0.44 A	14.2 ± 1.10 A	50.8 ± 2.10 A
*Mentha spicata*	6.2 ± 0.53 A	18.3 ± 0.89 A	49.6 ± 1.87 A
*Mentha piperita*	9.8 ± 0.23 A	13.4 ± 1.12 A	48.2 ± 2.05 A
*Mentha citrata*	8.7 ± 0.14 A	11.3 ± 0.98 A	55.3 ± 1.64 A
*Melaleuca alternifolia*	9.1 ± 0.23 A	12.4 ± 1.14 A	48.6 ± 0.85 A
*Ocimum basilicum*	8.3 ± 0.44 A	15.7 ± 2.02 A	49.3 ± 2.74 A
*Pogostemon cablin*	7.8 ± 0.46 A	13.5 ± 1.14 A	50.1 ± 1.76 A
Delegate 250 WG™	29.6 ± 0.34 B	35.7 ± 2.68 B	47.2 ± 2.30 A
F	6.5 ± 0.11	10.1 ± 1.74	46.2 ± 1.78
df	7.1 ± 0.23	11.5 ± 0.98	45.3 ± 2.35
*p*	119.11	78.45	11.32

^a^ Means followed by followed by different letters (A or B) on the columns of each figure indicate significant differences between treatments (one-way ANOVA, Dunnett test, *p* < 0.05). Data are presented as the mean ± standard error. Each experiment was performed 2 times with 10 replicates per treatment.

## Data Availability

Not applicable.

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
