# Peer review of "Essential Oils as a Source of Ecofriendly Insecticides for Drosophila suzukii (Diptera: Drosophilidae) and Their Potential Non-Target Effects"

_molecules, 2022, doi:10.3390/molecules27196215_

Round 1

Reviewer 1 Report

Dear Authors,

The submitted manuscript has high quality and I only have some minor suggestions for you:

1: Table 1 should be revised, as currently, it does not help the reader to identify, which essential oils share common chemicals. Also, the total percentage of the other, minor components could be added after each line, as there are some EOs with only two components (E. globulus) and ones with higher amount of minor components (O. basilicum)

2: Please clarify the positive control for each experiment. The marketed formula Delegate 250WG with 250 g/Kg spinetoram concentration is spinosad or are they two different chemicals? Try to use the same term in each figure and table as spinetoram is sometimes called espinetoram and more examples can be found.

3: What was the purity grades of the EOs? Were there any which met any the regulations pharmacopoeia? They were labelled as EO >90% purity or something like that?

4: Figure 1 could be modified with a simple ANOVA  and Dunnett's post hoc test, to compare each EO with Spinosad. As such it would be easier for the reader to compare the different plants. Optionally, I would consider dividing the Figure 1 into Figure 1 A and 1 B, as it seems too crowded. 

5: For the other statistical analysis I would also recommend using Dunnett's post hoc test to compare the treatments to the positive control.

6:  The subscripts marked with * should be added to the given figure's/table's description. Like Figure 2's "* Values are means obtained after..."

7: Check the manuscript again for mispellings like acetona (line 153).

Author Response

Dear reviewer,

We appreciate your fine work in the corrections.

1: Table 1 should be revised, as currently, it does not help the reader to identify, which essential oils share common chemicals. Also, the total percentage of the other, minor components could be added after each line, as there are some EOs with only two components (E. globulus) and ones with higher amount of minor components (O. basilicum)

Reply:  We appreciate your recommendation. Minority constituents added to the Table 1, please check L98-112.

2: Please clarify the positive control for each experiment. The marketed formula Delegate 250WG with 250 g/Kg spinetoram concentration is spinosad or are they two different chemicals? Try to use the same term in each figure and table as spinetoram is sometimes called espinetoram and more examples can be found.

Reply: Correct, we performed the standardization of substance for spinetoram throughout the manuscript.

3: What was the purity grades of the EOs? Were there any which met any the regulations pharmacopoeia? They were labelled as EO >90% purity or something like that?

Reply: Added sentence, L319-320: “All samples were standardized according to the methods and quality indicators described in the Brazilian Pharmacopoeia (≥ 90% purity).”

4: Figure 1 could be modified with a simple ANOVA  and Dunnett's post hoc test, to compare each EO with Spinosad. As such it would be easier for the reader to compare the different plants. Optionally, I would consider dividing the Figure 1 into Figure 1 A and 1 B, as it seems too crowded.

Reply: We appreciate your recommendation and changes have been made to figures 1, please check L121-125.

5: For the other statistical analysis I would also recommend using Dunnett's post hoc test to compare the treatments to the positive control.

Reply: Correct, we modified the analyzes in Figure 1, Table 4 (and 5, please check L L121-125, L179-182, and L205-210, respectively.

6:  The subscripts marked with * should be added to the given figure's/table's description. Like Figure 2's "* Values are means obtained after..."

Reply: The authors chose to remove the asterisks from the figure's/table's description.

7: Check the manuscript again for mispellings like acetona (line 153).

Reply: Correct, we performed the standardization 'acetone' in the text.

Reviewer 2 Report

Review report

The present research work reports the use of essential oils as an alternative in the fight against Drosophila suzukii. The authors have done important work to demonstrate that essential oils are capable of eliminating the insect. This work is important for developing ecofriendly means insecticides. However, it requires revision and additional information. Below are some comments that will improve the quality of this manuscript. 

- The document contains many missuses of English language

- Are these essential oils tested for the first time against this insect?

- What is the essential oil yield for each tested plant? can the authors qualify essential oils as eco-friendly if the yield is low?!! (we do not have the right to degrade an entire biodiversity to treat a single fruit).

- According to the authors : EOs are selected based on their commercial availability and chemical composition. Was the GC-MS analysis carried out before the selection of the plants or after?

- The authors are invited to redraw figure 1 to make it clear

- The authors must mention in table 1 the retention index of each product identified.

- Why the authors did not talk about the minor compounds since for some authors these compounds have a synergistic effect between them.

- Please formulate specific conclusion section,  the conclusions should be a response to the aim of the study.

Author Response

Dear reviewer,

We appreciate your fine work in the corrections.

- Are these essential oils tested for the first time against this insect?

Reply: Added information, L454-456: “In particular, this is the first report of the use of essential oil of C. martinii, C. flexuosus, E. staigeriana, M. cardiaca, M. spicata and M. citrata for the control of D. suzukii, as well as the first record for T. anastrephae.”

- What is the essential oil yield for each tested plant? can the authors qualify essential oils as eco-friendly if the yield is low?!! (we do not have the right to degrade an entire biodiversity to treat a single fruit).

Reply: As the EOs were acquired commercially, it is not possible to obtain the yield, since we did not extract and quantification of the samples. These plants have protocols for extracting EOs and domestication the species, as they are already commercialized.

- According to the authors: EOs are selected based on their commercial availability and chemical composition. Was the GC-MS analysis carried out before the selection of the plants or after?

Reply: Rewritten sentence, L318-320: “were selected based on their commercial availability. All samples were standardized according to the methods and quality indicators described in the Brazilian Pharmacopoeia (≥ 90% purity).” The GC-MS analysis carried out after obtaining the EOs.

- The authors are invited to redraw figure 1 to make it clear

Reply: Suggestion accepted, please check L98-112.

- The authors must mention in table 1 the retention index of each product identified.

Reply: Added information, please check L98-112.

- Why the authors did not talk about the minor compounds since for some authors these compounds have a synergistic effect between them.

Reply: Suggestion accepted, we added in Table 1 the minority constituents, please check L98-112. We also add, L225-228: “Likewise, the interactions of minority constituents contained in EOs have been reported to have synergistic action, providing a significant increase in the effectiveness of formulations for arthropod pests [26].”

- Please formulate specific conclusion section,  the conclusions should be a response to the aim of the study.

Reply: We appreciate your recommendation. Conclusions included, please check L448-461: “Chromatographic analysis of the 19 EOs samples used in this study revealed that the common components most frequently found were (E)-cinnamaldehyde and (E)-cinnamyl acetate in C. verum; α-pinene in C. sempervirens; geranial, neral, and geraniol in Cymbopgnon spp.; menthol and menthone in Mentha spp.; 1,8-cineole, geranial, and geraniol in Eucalyptus spp.; terpinen-4-ol in M. alternifolia; estragole in O. basilicum; and α-bulnesene in P. cablin. In particular, this is the first report of the use of essential oil of C. martinii, C. flexuosus, E. staigeriana, M. cardiaca, M. spicata and M. citrata for the control of D. suzukii, as well as the first record for T. anastrephae. Herein, EOs of C. verum, C. citratus QT citratus, and C. winterianus showed the highest toxicity in the ingestion bioassay for D. suzukii. The dry residues of C. verum and C. citratus QT citratus reduced the oviposition of D. suzukii and the 19 EOs analyzed repelled of females of D. suzukii. Interestingly, these EOs provided low mortality for T. anastrephae and did not cause reduction to the parasitism of T. anastrephae surviving females.”

Reviewer 3 Report

General Comments.- I found the paper interesting, especially the results with the parasitoid. It is interesting that the parasitoid is not susceptible to the EOs. This certainly warrants additional research.

Table 1. You might include the common names of the plants when possible. This would assist the reader not familiar with all these plants.

Fig. 1. The legend needs some clarification. I assume the Cap letters on each bar are the spray toxicity and the lower-case letters are ingestion. What does the asterisk refer to?

Specific Comments.

Line 53.    Since we really don’t understand the mode of action of many of the EO, I would think a contact spray would be toxic to parasitoids and predators. I would think you would need to test the EO against them as you have done in this paper.

Line 64-65.   I think the word “vapor activity” is actually more appropriate than fumigant activity. Essential oils are not fumigants. Fumigants such as methyl bromide and phostoxin act at the molecular level and are true fumigants.

Line 67 - Omit the word larvae.

Line 159-  Italics on Drosophila suzukii.

Lines 167-170. I think there is a Table that is missing. Table 5 shows the oviposition behavior of D. suzukii not the activity of EOs on the parasitoid. This is a really interesting finding. I just don’t understand why the EOs would be so selective. You might explain the lack of toxicity of EOs if the insect is exposed to them or feeding on the plant. In this case, the parasitoid is feeding on the host.

Line 238.  Both the oviposition test and the olfactory test were conducted on fresh deposits of the EO. Since these are typically short-lived compounds, would there be any long-term protective effects?

Lines 263-268. You mention in lines 48-49 that populations of D. suzukii have developed resistance to spinosad. What is the status of the population that you collected? It seems like having a population that is truly susceptible would be important. You mention breeding field populations annually with the lab colony. This might introduce resistance.

Lines 304-313. When I think of topical applications, a precise dose is applied to the insect with a micro applicator. The amount of toxicant per insect can be determined.  In this study it is actually a contact spray and you are correctly relating the data as a concentration.

Lines 315-321. I am not sure I understand this bioassay. Pieces of cotton strand are treated with the EOs. Do the flies drink the liquid from the cotton? Do the flies alight on the cotton? How do you eliminate the contact effects? How do you know that the flies imbibe the fluid?

Author Response

Dear reviewer,

We appreciate your fine work in the corrections.

Table 1. You might include the common names of the plants when possible. This would assist the reader not familiar with all these plants.

Reply: We appreciate your recommendation, please check L101-109: “Species: 1. Cinnamomum verum (cinnamon), 2. Cupressus sempervirens (cypress), 3. Cymbopogon citratus QT citratus (lemongrass), 4. Cymbopogon martini (palmarosa), 5. Cymbopogon flexuosus (lemongrass), 6. Cymbopognon citratus QT myrcene (lemongrass), 7. Cymbopognon winterianus (citronella grass), 8. Eucalyptus globulus (blue gum), 9. Eucalyptus radiata (Forth River peppermint), 10. Eucaliptus staigeriana (lemon Ironbark), 11. Eucalyptus citriodora (citriodora), 12. Mentha arvensis (wildmint), 13. Mentha cardiaca (gingermint), 14. Mentha spicata (spearmint), 15.  Mentha piperita (peppermint), 16. Mentha citrata (eau de cologne mint), 17. Melaleuca alternifolia (tea tree), 18. Ocimum basilicum (basil) and 19. Pogostemon cablin (patchouli)”.

Fig. 1. The legend needs some clarification. I assume the Cap letters on each bar are the spray toxicity and the lower-case letters are ingestion. What does the asterisk refer to?

 Reply: We made changes to Figure 1 and legend, please check L121-125: “Data are presented as the mean ± standard error. Means followed by followed by different letters on the columns of each figure indicate significant differences between treatments (one-way ANOVA, Dunnett test, P < 0.05).”

Specific Comments.

Line 53.    Since we really don’t understand the mode of action of many of the EO, I would think a contact spray would be toxic to parasitoids and predators. I would think you would need to test the EO against them as you have done in this paper.

Reply: We performed the proposed test. Please check Table 5 (L205-210).

Line 64-65.   I think the word “vapor activity” is actually more appropriate than fumigant activity. Essential oils are not fumigants. Fumigants such as methyl bromide and phostoxin act at the molecular level and are true fumigants.

Reply: Suggestion accepted, please check L64-65: “was shown to cause a vapor activity on adults”

 Line 67 - Omit the word larvae.

Reply: The authors removed the word larvae.

Line 159-  Italics on Drosophila suzukii.

Reply: Correct, changes made (L179).

Lines 167-170. I think there is a Table that is missing. Table 5 shows the oviposition behavior of D. suzukii not the activity of EOs on the parasitoid. This is a really interesting finding. I just don’t understand why the EOs would be so selective. You might explain the lack of toxicity of EOs if the insect is exposed to them or feeding on the plant. In this case, the parasitoid is feeding on the host.

Reply: Correct, please check L205-210:  Table 5. Mortality and parasitism of Trichopria anastrephae at 120 h after exposure to treatments in ingestion and topical application bioassays.

Line 238.  Both the oviposition test and the olfactory test were conducted on fresh deposits of the EO. Since these are typically short-lived compounds, would there be any long-term protective effects?

Reply: Yes, we also demonstrated that the flies showed mortality by the topical method and ingestion up to 120 h after exposure to the treatments.

Lines 263-268. You mention in lines 48-49 that populations of D. suzukii have developed resistance to spinosad. What is the status of the population that you collected? It seems like having a population that is truly susceptible would be important. You mention breeding field populations annually with the lab colony. This might introduce resistance.

Reply: Susceptible, added information L312-315: “Both D. suzukii and T. anastrephae are colonies susceptible to chemical insecticides”. The introduced insects are obtained from organic greenhouse without the application of chemicals.”

Lines 304-313. When I think of topical applications, a precise dose is applied to the insect with a micro applicator. The amount of toxicant per insect can be determined.  In this study it is actually a contact spray and you are correctly relating the data as a concentration.

Reply: The authors chose to work with the application of spray concentrations to get closer to field trials.

Lines 315-321. I am not sure I understand this bioassay. Pieces of cotton strand are treated with the EOs. Do the flies drink the liquid from the cotton? Do the flies alight on the cotton? How do you eliminate the contact effects? How do you know that the flies imbibe the fluid?

Reply:  Excellent notes.

Do the flies drink the liquid from the cotton? Although, we have not quantified the consumption of EOs by flies, we observed the insects sucking the solution.

Do the flies alight on the cotton? Yes.

How do you eliminate the contact effects? Does not eliminate the contact effects. However, the contact effect is "less pronounced" than with topical application, as flies take longer to die in the ingestion bioassay.